# Methicillin Resistance Elements in the Canine Pathogen *Staphylococcus pseudintermedius* and Their Association with the Peptide Toxin PSM-mec

**DOI:** 10.3390/antibiotics13020130

**Published:** 2024-01-28

**Authors:** Gordon Y. C. Cheung, Ji Hyun Lee, Ryan Liu, Sara D. Lawhon, Ching Yang, Michael Otto

**Affiliations:** 1Pathogen Molecular Genetics Section, Laboratory of Bacteriology, National Institute of Allergy and Infectious Diseases (NIAID), US National Institutes of Health (NIH), Bethesda, MD 20892, USA; cheunggo@niaid.nih.gov (G.Y.C.C.); christina.lee2@nih.gov (J.H.L.); ryan.liu@nih.gov (R.L.); 2Department of Veterinary Pathobiology, College of Veterinary Medicine and Biomedical Sciences, Texas A&M University, College Station, TX 77843, USA; slawhon@cvm.tamu.edu; 3Department of Veterinary Biomedical Sciences, College of Veterinary Medicine, Long Island University, Brookville, NY 11548, USA; ching.yang@liu.edu

**Keywords:** *Staphylococcus pseudintermedius*, phenol-soluble modulin, PSM-mec, SCC*mec*, Agr, CRISPR, restriction-modification system

## Abstract

*Staphylococcus pseudintermedius* is a frequent cause of infections in dogs. Infectious isolates of this coagulase-positive staphylococcal species are often methicillin- and multidrug-resistant, which complicates therapy. In staphylococci, methicillin resistance is encoded by determinants found on mobile genetic elements called Staphylococcal Chromosome Cassette *mec* (SCC*mec*), which, in addition to methicillin resistance factors, sometimes encode additional genes, such as further resistance factors and, rarely, virulence determinants. In this study, we analyzed SCC*mec* in a collection of infectious methicillin-resistant *S. pseudintermedius* (MRSP) isolates from predominant lineages in the United States. We found that several lineages characteristically have specific types of SCC*mec* elements and Agr types and harbor additional factors in their SCC*mec* elements that may promote virulence or affect DNA uptake. All isolates had SCC*mec*-encoded restriction–modification (R-M) systems of types I or II, and sequence types (STs) ST84 and ST64 had one type II and one type I R-M system, although the latter lacked a complete methylation enzyme gene. ST68 isolates also had an SCC*mec*-encoded CRISPR system. ST71 isolates had a *psm-mec* gene, which, in all but apparently Agr-dysfunctional isolates, produced a PSM-mec peptide toxin, albeit at relatively small amounts. This study gives detailed insight into the composition of SCC*mec* elements in infectious isolates of *S. pseudintermedius* and lays the genetic foundation for further efforts directed at elucidating the contribution of identified accessory SCC*mec* factors in impacting SCC*mec*-encoded and thus methicillin resistance-associated virulence and resistance to DNA uptake in this leading canine pathogen.

## 1. Introduction

*Staphylococcus pseudintermedius* is, like *Staphylococcus aureus* and in contrast to many other non-*S. aureus* staphylococci, a coagulase-positive *Staphylococcus* species. It is a member of the animal pathogen *Staphylococcus intermedius* group (SIG) [1]. *S. pseudintermedius* is considered a significant opportunistic pathogen of dogs [2], where it is responsible for causing a variety of diseases, such as pyoderma (characterized as skin and soft tissue infections that scale in severity), and infections of the urinary tract and external ear canal [3]. Dog-biting incidents of humans by colonized or infected animals [4] can lead to the zoonotic transmission of *S. pseudintermedius*, including methicillin-resistant *S. pseudintermedius* (MRSP), to humans [5,6]. Since the first reports of MRSP in 2005 [7], there has been a sharp rise in multidrug resistance amongst MRSP isolates [2], making routine treatment of the aforementioned diseases much more challenging for dogs and their owners [8]. The factors that underlie the success of some *S. pseudintermedius* lineages as canine pathogens remain mostly unknown.

Undisputedly, resistance to methicillin (a beta-lactam antibiotic) is the most important antibiotic resistance determinant in staphylococci and is linked to the acquisition of a mobile genetic element called Staphylococcal Chromosome Cassette *mec* (SCC*mec*). Resistance to methicillin is afforded by the production of the low-affinity penicillin binding protein 2A (PBP2A) [9], encoded by the *mecA* gene found within the *mec* gene complex, one of two defining hallmark composites of the SCC*mec* element. The second composite is the cassette chromosome recombinase (*ccr*) gene complex, which harbors *ccr* genes responsible for the integration of the element. The *mec* and *ccr* gene complexes are separated by three non-essential J (formally known as junkyard) regions [10] that harbor other antibiotic and metal resistance genes. A classification system originally established to distinguish *S. aureus* SCC*mec* elements is based on permutations, gene variants, and the presence or absence of genes [11,12]. However, the recent molecular typing and characterization of SCC*mec* elements from staphylococcal species other than *S. aureus* have reported exceedingly complex compositions that do not conform to the original typing scheme, for which *S. pseudintermedius* is an excellent example [13,14,15,16,17,18,19,20,21]. In such instances, the use of SCC*mec* followed by the name of the strain has been proposed [11].

The whole-genome sequencing (WGS) of several dominant lineages has provided important clues about the molecular epidemiology and biological characteristics of *S. pseudintermedius* [22,23]. Currently, more than half of all isolates collected in the United States belong to sequence type (ST) 71, which is also expanding globally [20]. Although ST71 is dominant in the US, several other *S. pseudintermedius* lineages are also prevalent, including ST64, ST68, and ST84 [24]. 

Similar to its pathogenic human cousin, *S. aureus*, *S. pseudintermedius* harbors an impressive number of genes that encode a variety of virulence determinants, such as immune-evasion proteins [2,25] and a plethora of toxins including an exfoliative toxin [26,27], a bicomponent leucocidin [28], a superantigen [29], and several peptide toxins of the phenol-soluble modulin (PSM) superfamily [30]. PSMs possess similar alpha-helical and amphipathic secondary structures, despite lacking any significant amino acid identity [31], and play significant roles in the pathogenesis of *S. aureus* and the opportunistic pathogen *S. epidermidis*, mediated by their pro-inflammatory, cytolytic, and biofilm-structuring functions [32,33,34,35]. In many staphylococci, it has been shown that the regulation of most virulence determinants is dependent on the presence of the *agr* (accessory gene regulator) locus. This locus codes for a classical two-component signal transduction system and a regulatory RNA (RNAIII) that controls the transcription of virulence genes and often also contains the PSM δ-toxin gene [36,37]. 

First described in *S. aureus*, the *psm-mec* locus encoding the methicillin resistance-associated PSM, PSM-mec, is found on the class A *mec* gene complex of specific SCC*mec* elements (belonging to types II, III, and VIII) of many staphylococcal species [38,39,40]. The *psm-mec* gene is embedded in a DNA sequence coding for a small regulatory (sr) RNA, next to the *mecI*/*mecR1*/*mecA* genes [41,42] and, as demonstrated in *S. aureus*, is under RNA-independent control by Agr like the other PSMs [43,44]. To date, the contribution of the *psm-mec* locus towards virulence has only been investigated in *S. aureus* and *S. epidermidis*. In *S. aureus*, deletion of the *psm-mec* gene only impacts the virulence of isolates expressing greater levels of PSM-mec compared to other PSMs [40], indicating that PSM-mec may influence pathogenesis only in certain backgrounds. In contrast, *S. epidermidis* pathogenesis is strongly dependent on PSM-mec expression [45]. In addition to the likely importance of the relative production of PSM-mec as compared to other strongly cytolytic PSMs, these differences may be a result of the additional level of transcriptional control by the *psm-mec* srRNA [42]. Genomic comparisons of different *S. pseudintermedius* lineages have revealed the presence of the *psm-mec* gene predominantly in SCC*mec* type III elements [13,38], but no studies have determined whether these isolates are capable of producing the peptide. In *S. pseudintermedius*, the type III SCC*mec* element is mostly associated with MRSP sequence type (ST) 71. 

In this study, we sought to characterize SCC*mec* elements from MRSP isolates representing the four major lineages within the United States in detail to further expand our current knowledge of what may explain their current expansion in North America. To that end, we also determined the production of PSM-mec in *S. pseudintermedius* and report that PSM-mec is expressed in specific methicillin-resistant isolates of that species.

## 2. Results

### 2.1. Selection of Isolates for SCCmec Characterization

In our previous study, 160 clinical *S. pseudintermedius* isolates from canines were whole-genome-sequenced [24] and their genomes were examined for the presence of antimicrobial resistance genes and the prevalence of virulence genes. As the problem of multidrug-resistant (MDR) *S. pseudintermedius* infections has become particularly prevalent in veterinary clinics, we sought to characterize MRSP isolates from this collection. These isolates were collected prior to 2013. Our previous work determined that the presence of the *mecA* gene varied between each ST/Agr group [24]. For instance, the *mecA* carriage rate for ST84/Agr group I, ST64/Agr group II, ST71/Agr group III, and ST68/Agr group IV were 45% (5/11), 18% (9/50), 34% (15/44), and 42% (23/55), respectively (Table 1); the total number of *mecA*-positive isolates was only 32%. 

For this study, we focused on a smaller number of MRSP isolates for SCC*mec* characterization and subsequent analysis of PSM-mec production using the following isolate selection process: We chose approximately 50% of the *mecA*-positive isolates for each ST/Agr group and in a manner in which each group contained at least one isolate from pyoderma, urinary tract, or surgical wound infections (Table 1). This resulted in the selection of 5-12 *mecA*-positive isolates from various sources for each group. As the number of MRSP isolates from ST84/Agr group I was small (n = 5), all five available isolates were selected for that group. Among the isolates, two were from healthy animals. Isolates 37-032, 32-012, and the whole-genome-sequenced strain, ED99, were chosen as methicillin-sensitive (MSSP) controls. In summary, we included 29 MRSP isolates and three methicillin-sensitive *S. pseudintermedius* (MSSP) isolates in this study. As expected, all MRSP isolates demonstrated phenotypic resistance to oxacillin (>0.25 µg/mL) and penicillin (≥8 µg/mL) as evidenced by MIC (Table 1), while isolates 37-032, 32-012, and ED99 were oxacillin-sensitive.

### 2.2. Characterization of SCCmec Elements

The prediction and characterization of SCC*mec* types in our selected cohort of 29 MRSP isolates [24] were initiated with the online resource, SCC*mec*Finder 1.2, (https://cge.food.dtu.dk/services/SCCmecFinder-1.2/, accessed on 27 September 2023) [46]. SCC*mec*Finder determined that all the submitted MRSP WGS sequences contained the *mecA* gene. However, it was only able to assign an SCC*mec* type for six out of seven ST71/Agr group III isolates and 11 out of 12 ST68/Agr group IV isolates (Table 1). A closer inspection of the whole-genome sequences of the non-typable isolates showed that the genes associated with the *mec* and *ccr* markers were spread across multiple contigs. Short-read sequencing technology, which was used at the time, often presents problems with full assembly of the entire SCC*mec* sequence, most likely mediated by multiple insertion sequences [11]. Therefore, we performed long-read WGS for the remaining non-typable isolates. Genomic analyses of the SCC*mec* elements showed that there was only one dominant SCC*mec* type in each ST/Agr group. Further details are described below.

#### 2.2.1. ST84/Agr Group I and ST64/Agr Group II

All five isolates from the ST84/Agr group I and one out of five ST64/Agr group II isolates showed a genetic layout that was identical to that of a ~52 kb SCC*mec* element recently described in an *S. pseudintermedius* isolate (NA45) from Tennessee, United States (accession no; NZ_CP016072) [23] (Figure 1). A Clustal Omega nucleotide alignment of the SCC*mec* elements from these six MRSP isolates with SCC*mec*_NA45_ revealed that they were 99% identical. Interestingly, the *mec* gene complex of SCC*mec*_NA45_ is found in the opposite orientation compared to other SCC*mec* elements. SCC*mec*_NA45_ harbors *ccrC1* allele 6, which is 100% homologous to the same gene found in *S. haemolyticus* isolate 25–60 [47]. Additionally, SCC*mec*_NA45_ harbors cadmium- (*cadA*) and arsenic- (*arsC*, *arsB,* and *arsR*) resistance genes and genes belonging to Type-I and Type-II restriction–modification (R-M) systems. Interestingly, the Type-I R-M system comprises three genes, *hsdM*, *hsdR,* and *hsdS*, but, in these isolates, *hsdM* includes a premature stop codon, leading to gene truncation. In contrast, full copies of the genes of the Type-II R-M system are present. The genomic layouts for the four remaining ST64/Agr group II isolates (11-041, 29-036, 30-027, 42-072) are identical to those of SCC*mec*_NA45_ with the exception that a 2.2 kb region harboring a pair of integrase genes, *intA* and *intB* (Figure 1), is found slightly downstream of *ccrC1* allele 6. We refer to this SCC*mec* element as SCC*mec*_NA45_*^int^*.

#### 2.2.2. ST71/Agr Group III

The SCC*mec* regions of all ST71/Agr group III isolates displayed the same genetic arrangement of genes (Figure 1). SCC*mec*Finder predicted the prototype class A *mec* gene complex, which contains *mecA* and full nucleotide sequences of *mecRI* and *mecI*, and the *ccr* gene complex 3, which is represented by the recombinase genes *ccrA3* and *ccrB3*. According to whole-genome analysis and blastn searches, this combination had 99% sequence identity to an SCC*mec* element originally referred to as SCC*mec* Type II-III hybrid (accession no. AM904732) [48], but has recently been suggested to be renamed to SCC*mec* III_(KM1381)_, owing to its unique genetic characteristics [13,18]. SCC*mec* III_(KM1381)_ has a length of ~39 kb, approximately 13 kb shorter than SCC*mec*_NA45_. Major features of this SCC*mec* element include a type I restriction–modification (RM) system, which has not been described previously, and the *psm-mec* locus. In contrast to the *hsd* locus of ST84/Agr group I and ST64/Agr group II, the full *hsdM*, *hsdR*, and *hsdS* genes of ST71/Agr group III are found (in the opposite orientation).

On the other hand, the *psm-mec* locus, which comprises the *psm-mec* gene embedded in a small regulatory (sr)RNA, was found adjacent to the class A *mec* gene complex, sandwiched between *mecI* and *mecR2* (*xylR*) [42,43]. Based on the fact that the *psm-mec* locus is associated with SCC*mec* types II, IIA, IIB, IID, III, and VIII of multiple *Staphylococcus* species [38,39], it was reasonable to assume that only ST71/Agr group III harboring SCC*mec* III_(KM1381)_ harbored that locus. Indeed, we found that the *psm-mec* gene and the *psm-mec* srRNA were exclusive to this ST/Agr group (Figure 1, Table 1) and had 100% sequence identity to those described in *S. aureus* and *S. epidermidis* [49]. A characterized -7 T > C mutation in the *psm-mec* promoter region, which attenuates PSM-mec expression, is present in a subset of hospital-associated methicillin-resistant *S. aureus* (HA-MRSA) strains harboring SCC*mec* Type II [41,50,51,52]. However, the *psm-mec* promoter region in our ST71/Agr group III isolates did not harbor this mutation, indicating host-species-independent specificity for SCC*mec* Type II elements.

#### 2.2.3. ST68/Agr Group IV

All 12 sequenced isolates belonging to the ST68/Agr group IV harbored the same type of SCC*mec* element carrying the class C2 *mec* gene complex flanked by two *ccrC1* genes (alleles 2 and 8) (Figure 1 and Figure 2). The *mec* and *ccr* gene complexes of this large 56 kb element are mostly homologous to *S. aureus* SCC*mec* Vb [11], formerly known as SCC*mec* type V (5C2&5)/SCC*mec* type V_T_/VII from *S. aureus* (accession no. AB462393) [53,54]. However, compared to the *S. aureus* SCC*mec* sequence, significant disparities were found downstream of the *ccrC1* allele 2 gene. For instance, remains of a Type I R-M system (i.e., a truncated *hsdR* gene and no *hsdM* or *hsdS* genes), followed by a Type IIIA CRISPR-Cas complex, genes for a Type II R-M system (with a truncated methylase gene), and a heavy metal resistance gene (Figure 1 and Figure 2) were observed in the ST68/Agr group IV SCC*mec* elements. The entire SCC*mec* element showed an architecture similar to that of recently published whole-genome sequences of V_T_ isolates from Australia and Argentina [16,18], with a 98–99% sequence identity to SCC*mec* V_T_ of an isolate from Ireland (ERR175868) [55] and the South Korean isolate Z0118SP0108 (CP061030.1). Variances in the number of direct repeats (DRs) (n = 5–18) found in CRISPR array 1, as well as the presence or absence of a second CRISPR array (always with 4 DRs) located immediately downstream of *cas6* in some isolates, accounted for sequence differences. 

### 2.3. Detection of PSM-mec in Culture Filtrates

The presence of the *psm-mec* locus in *S. pseudintermedius* is generally overlooked and, according to our knowledge, there are no available reports describing PSM-*mec* protein production in this species. Therefore, we analyzed 16-h culture filtrates from our isolate collection using an established RP-HPLC/ESI-MS method [31]. As expected, PSM-*mec* could not be detected in filtrates of MSSP isolates lacking an SCC*mec* element (37-032, 32-012, and ED99) or any isolate lacking the *psm-mec* locus (ST84/Agr group I, ST64/Agr group II, and ST68/Agr group IV) (Table 1; Figure 3). Notably, PSM-*mec* production was exclusive to the ST71/Agr III group that harbored SCC*mec* III_(KM1381)_, which correlates with the presence of the *psm-mec* locus from our WGS data analyses (Table 1; Figure 3). The production levels of PSM-mec in these isolates from the ST71/Agr III group ranged from 5 to 27 μM, which is in the range of production in the *S. epidermidis* isolate SE620, while the clinical *S. aureus* MSA3407, a documented high producer of PSM-mec, produces more than 20-fold more PSM-mec than all the *S. pseudintermedius* isolates. Five of the seven *psm-mec*-gene-positive isolates were PSM-*mec* producers. The two *psm-mec*-gene-positive isolates that did not show PSM-mec production were also δ-toxin-negative, which is indicative of a dysfunctional Agr system (Table 1) [56]. The absence of PSM-mec production in those isolates is thus expected, assuming that Agr controls *psm-mec* expression in *S. pseudintermedius* as it does in *S. aureus* [43].

## 3. Discussion

Comprehensive analyses of the accessory genomes, including specifically methicillin resistance determinants of *S. pseudintermedius* lineages, have been published before [13,17,20]. We focused here on the most widely distributed lineages associated with canine infections in the United States and the analysis of factors potentially determining pathogenic success, including particularly PSM-mec.

In our isolate cohort, ST was strictly associated with Agr type and SCC*mec* type. As we reported in our previous study [24], we did not find any association between disease type and Agr type, which contrasts findings in *S. aureus* [57]. There was also no association between disease type and SCC*mec* type in our isolates.

While the presence of the *psm-mec* gene in two isolates of *S. pseudintermedius* has been reported [38], there has been no analysis of its presence in a larger cohort of isolates and no analysis of its expression as a secreted peptide toxin. We found that the presence of *psm-mec* was associated with ST71/Agr group III, and SCC*mec* III_(KM1381)_ in MRSP, which corresponds to the two isolates analyzed by Monecke et al. [38]. Notably, all isolates of that group had the *psm-mec* locus, which revealed a 100% nucleotide sequence identity to those found in *S. aureus* strain 252 and *S. epidermidis* strain RP62A. Not all these isolates also produced the PSM-mec peptide. Because the non-producers were also δ-toxin-negative, this is likely due to Agr dysfunctionality. The widespread occurrence of dysfunctional Agr mutants has repeatedly been reported in staphylococci [58,59,60]. PSM-mec production was much lower than in clinical MRSA. In *S. aureus*, 25% of clinical MRSA was found to have a T to C mutation at position -7 relative to the start codon in the *psm-mec* promoter that suppresses translation [41,51]. No *psm-mec*-positive MRSP isolates in this study harbored this mutation, suggesting that other, potentially species-specific factors control PSM-mec production in *S. pseudintermedius* and are responsible for the comparatively low levels found in culture filtrates. The contribution of PSM-mec and the *psm-mec* RNA to MRSP virulence will need to be investigated in the future using site-specific allelic replacement mutants. Furthermore, it is possible that PSM-mec may play a role in the observed recent spread of ST71 in the United States and elsewhere [20].

The SCC*mec* elements of the MSRP isolates analyzed in this study did not encode any virulence determinants other than PSM-mec. However, there were differences in genes and gene clusters between them that may affect bacterial survival and genetic adaptation. Namely, some isolates harbored genes encoding type I and/or type II R-M systems in complete arrays or with truncations, while others had CRISPR loci. ST64 and ST84 isolates had both a type I and a type II R-M system.

R–M systems are commonly found in prokaryotic genomes and protect against foreign DNA invasion. Three subunits are required for type I R-M functionality, HsdS, HsdM and HsdR, which are responsible for recognition of a specific sequence, methylation to protect from self-digestion, and an endonuclease to cleave the non-protected DNA, respectively [61]. Interestingly, the type I R-M system appears to have a truncated *hsdM* gene in ST64 and ST84 isolates. It remains to be analyzed whether this situation affects the functionality of the system and self-digestion. Two separate proteins are needed for the type II R-M system [62,63] and in all ST64 and ST84 isolates, the genes encode full proteins. However, the methylase gene in ST68 isolates was truncated.

CRISPR (Clustered Regularly Interspaced Short Palindromic Repeats) are bacteriophage-derived DNA sequences in prokaryotic genomes that originate from previous bacteriophage infection. CRISPR-Cas systems are used to digest DNA from similar bacteriophages during subsequent infections. They may limit the exchange of genetic material between bacteria, including, for example, antimicrobial resistance determinants [64]. CRISPR sequences have been detected in *S. pseudintermedius*, sometimes associated with SCC*mec* elements, and used as markers to analyze the spread of antimicrobial resistance among the staphylococci [65]. Whether the CRISPR-Cas system encoded on the type V SCC*mec* elements of MSRP ST68 limits the uptake of foreign DNA and impacts survival or genetic adaptation remains to be investigated.

Finally, in some ST64/Agr group II isolates, we noted the presence of a pair of integrase genes, *intA* and *intB,* in the SCC*mec* elements. These integrases may be important for the mobility of pathogenicity islands in bacterial genomes and thus for bacterial evolution [66]. However, we did not find pathogenicity islands in our isolates such as the one observed in MRSP ST181 [67].

## 4. Materials and Methods

### 4.1. Bacterial Strains and Culture

In this study, 29 methicillin-resistant *S. pseudintermedius* (MRSP) and three methicillin-sensitive *S. pseudintermedius* clinical isolates were selected from the Texas Veterinary Medical Teaching Hospital strain collection at Texas A&M University between 2007 and 2016 [24]. One MSSP isolate (36-033) was used as a negative control for *mecA* detection; the other two isolates (37-032 and 32-012) were chosen for WGS analyses. At the time of isolation, bacteria were identified to the level of the *S. intermedius* group using biochemical tests including coagulase, urease, the ability to utilize trehalose and mannitol salt, and later confirmed via matrix-assisted laser desorption ionization time-of-flight mass spectrometry (MALDI-TOF MS; Biotyper, Bruker, Billerica, USA). Identification to the species level was performed using PCR as previously described [68]. The genome-sequenced MSSP type strain, ED99, and the methicillin-resistant *S. aureus* isolate MSA3407 were from Fitzgerald et al. [22,69,70]. Strain SE620 is a methicillin-resistant *S. epidermidis* clinical isolate from Norway [71]. *S. aureus* ATCC 29213 and ATCC 43300, *E. faecalis* ATCC 29212, *E. coli* ATCC 25922, and *P. aeruginosa* ATCC 27853 were acquired from American Type Culture Collection (ATCC). All bacteria were streaked from glycerol stocks onto Tryptic soy agar (TSA) or TSA supplemented with 5% sheep blood (BD) and incubated overnight at 37 °C to yield single colonies. A colony was selected and inoculated into tryptic soy broth (TSB) and grown for 16 h with shaking at 180 rpm at 37 °C, unless stated otherwise, for subsequent experiments. 

### 4.2. Minimal Inhibitory Concentration (MIC) Assay

A commercially available antimicrobial drug susceptibility test panel (CompGP1F; Sensititre; Thermo Fisher Scientific, Waltham, MA, USA) was used to determine the minimum inhibitory concentration (MIC) of antimicrobial drugs for the *S. pseudintermedius* isolates from canines. Testing was performed according to the Clinical and Laboratory Standards Institute (CLSI) Guidelines [72]. Quality control testing for the laboratory consisted of weekly testing of the microbroth dilution tests using *Staphylococcus aureus* ATCC 29213, *Enterococcus faecalis* ATCC 29212, *E. coli* ATCC 25922, and *Pseudomonas aeruginosa* ATCC 27853 in accordance with the CLSI guidelines published in the document, VET01S.

### 4.3. DNA Isolation, Sequencing, Genome Assembly, and Alignment

*S. pseudintermedius* isolates were washed with phosphate-buffered saline, pelleted, adjusted to the desired concentration, and resuspended in 0.5 mL of 1× DNA/RNA Shield (Zymo Research, Irvine, CA, USA) to lyse bacteria and stabilize nucleic acids. Samples were sent to Plasmidsaurus (Eugene, OR, USA) where genomic DNA was isolated, minimally fragmented, and prepared for long-read sequencing using Oxford Nanopore Technologies (ONT, Oxford, United Kingdom). Amplification-free libraries were constructed from the fragmented DNA using the Kit v14 library prep chemistry, and the libraries were sequenced using R10.4.1 flow cells (ONT). The flow cells comprise an electro-resistant membrane harboring an array of nanopores, each individually connected to a separate electrode. As molecules pass through the pore, unique changes in ionic current are detected by associated channel and sensor chips. These individual signatures are interpreted by algorithms, which then generate DNA sequence data in real-time. High-quality consensus sequences were generated by Plasmidsaurus through data processing with Flye (Version 2.9.1) and Medaka (Version 1.8.0) for assembly and polishing, respectively, and lastly, bacterial genome annotations were produced with Bakta (Version 1.6.1).

### 4.4. Comparative Genomics

The SCC*mec*Finder 1.2, a database (https://cge.food.dtu.dk/services/SCCmecFinder-1.2/, accessed on 27 September 2023) [46] that contains information from *S. aureus* SCC*mec* types I through XI (at the time of manuscript preparation), was employed to predict SCC*mec* types from WGS data. For each isolate, additional analyses were performed by comparing the presence of genetic elements as described in the guidelines by the International Working Group on the Classification of Staphylococcal Cassette Chromosome (IWG-SCC) [12]. Online tools, CRISPRfinder and (https://crisprcas.i2bc.paris-saclay.fr/CrisprCasFinder/Index, accessed on 15 October 2023) [73] and IS finder https://isfinder.biotoul.fr (accessed on 15 October 2023) [74], were used to detect and characterize clustered regularly interspaced short palindromic repeat (CRISPR) elements and insertional sequences, respectively. Finally, manual alignment/mapping was carried out using the available reference sequences for *S. aureus* and *S. pseudintermedius* SCC*mec* types. The CLUSTAL Omega algorithm (Version 1.2.3) was used to align sequences in Geneious (Version 2023.0.4) and comparative genome figures were created.

### 4.5. Detection of mecA using Real-Time PCR

For genomic DNA isolation, one bacterial colony from a TSA plate was resuspended into 50 μL of nuclease-free water and heated at 95 °C for 10 min. After heating, the samples were centrifuged down for 10 min at 10,000× *g* and the DNA-containing pellet was used for real-time PCR for the identification of the *mecA* gene as described previously [75]. Briefly, primers, LTmecAF (5′-AAAGAACCTCTGCTCAACAAGT-3′) and LtmecAR (5′-TGTTATTTAACCCAATCATTGCTGTT-3′), and probe, LTmecAHP2 (5′-[6FAM]CCAGATTACAACTTCACCAGGTTCAAC[BHQ1]-3′), were diluted to final concentrations of 500 nM in PCR reaction volume of 25 μL consisting of TaqMan Fast Universal PCR Master Mix (2×) (Thermo Fisher Scientific), 2 μL DNA template, and water. Real-time PCR was performed on an Applied Biosystems 7500 Fast Thermocycler (Thermo Fisher Scientific) using the presence/absence assay setting with cycling parameters that consisted of an initial step of 60 °C for 1 min and 95 °C for 20 s, followed by 40 cycles of 95 °C for 3 s, and then 60 °C for 30 s with a final step at 60 °C for 1 min. The *mecA*-positive control was MRSA ATCC 43300. The negative controls included MSSA ATCC 29213 and MSSP isolate 36-033 that we previously characterized [24].

### 4.6. Analysis of PSM-mec Production by RP-HPLC/MS

For the identification and quantification of PSM-mec, culture filtrates were collected from 16 h cultures of all *S. pseudintermedius* isolates and reference strains (*S. pseudintermedius* ED99, *S. epidermidis* isolate SE620, *S. aureus* isolate MSA3407) and then subjected to reversed-phase high-pressure liquid chromatography/electrospray mass spectrometry (RP-HPLC/ESI-MS) as recently described [76]. For the absolute quantification of PSMs in culture filtrates, known molar concentrations of the PSM-mec peptide (MDFTGVITSIIDLIKTCIQAFG) synthesized with an N-terminal N-formyl methionine modification at >95% purity (Peptide 2.0) were made. Peptide was first dissolved into dimethyl sulfoxide (DMSO) to a stock concentration of 10 mg/mL and further diluted into water to generate standards. The simple linear regression function in GraphPad Prism software (Version 9.5.1) was used to calculate the PSM-mec concentrations. To account for differences in PSM production in batch runs performed on different days, one selected *S. pseudintermedius* isolate was always included as a reference and then the values were used for normalization. The RP-HPLC/ESI-MS method was also used to detect δ-toxin.

### 4.7. Data Availability

The WGS-annotated genomes containing contigs with SCC*mec* sequences are publicly available on the Bacterial And Viral Bioinformatics Resource Center website (https://www.bv-brc.org, accessed on 19 September 2023) under the name “Slittle_StaphPseudAGR” in the BV-BRC public workspaces portal [24]. Twelve isolates, whose SCC*mec* regions could not be identified, because sequences spanned multiple contigs, were re-sequenced, and the raw sequence data in *.fastq.gz format were deposited to the National Center for Biotechnology Information (NCBI) database under bioproject PRJNA1055088 with accession numbers SAMN38979297–SAMN38979308 (see Table 1).

## Figures and Tables

**Figure 1 antibiotics-13-00130-f001:**
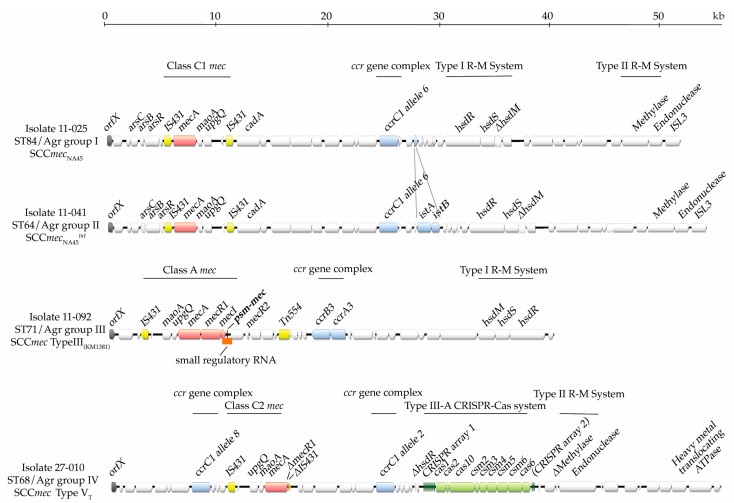
Genetic layout of SCC*mec* elements representative of ST84/Agr group I, ST64/Agr group II, ST71/Agr group III, and ST68/Agr group IV *S. pseudintermedius* isolates. A pair of integrase genes (*intA* and *intB*) are found in some ST64/Agr group II isolates. The number of direct repeats (DRs) in CRISPR array 1, represented by dark green arrows in the SCC*mec* elements of ST68/Agr group IV isolates, vary between 5 and 18. In some isolates, a second CRISPR array is found downstream of the CRISPR protein machinery (light green arrows). *orfX*, *mecA*, IS elements, and recombination proteins are depicted by gray, red, yellow, and blue arrows, respectively. The *psm-mec* locus, comprising the *psm-mec* gene (black arrow) and the srRNA (orange bar), is found exclusively in the SCC*mec* of ST71/Agr group III isolates. R-M, restriction–modification; ISL3, transposase.

**Figure 2 antibiotics-13-00130-f002:**
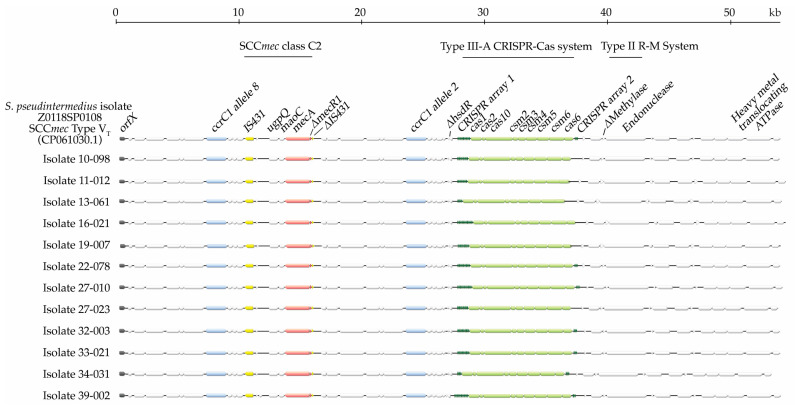
Alignment of *S. pseudintermedius* ST68/Agr group IV SCC*mec* elements with that of *S. pseudintermedius* isolate Z0118SP0108 (CP061030.1).

**Figure 3 antibiotics-13-00130-f003:**
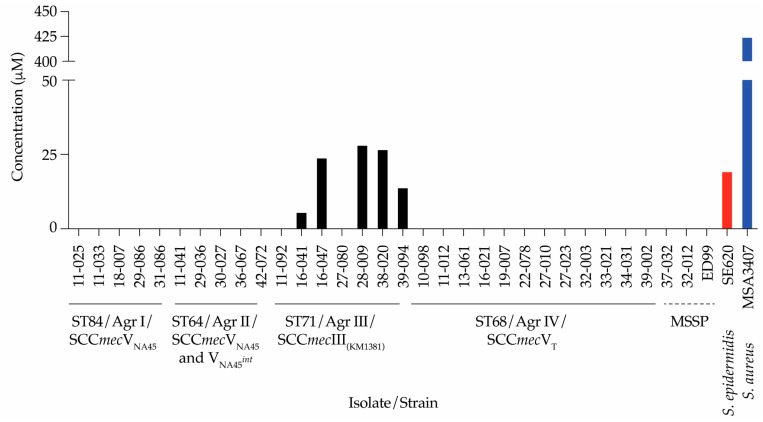
PSM-mec production in 16 h culture filtrates of a collection of MRSP and MSSP isolates. *S. epidermidis* SE620 and *S. aureus* MSA3407, previously described PSM-mec producers, are denoted by red and blue columns, respectively, for comparison.

**Table 1 antibiotics-13-00130-t001:** Characteristics of *S. pseudintermedius* canine isolates in this study.

Isolate Number	Disease Type	Source	Accession Number	MLST ^a^	Agr Group ^b^	Collection Date	*mecA* PCR ^c^	Oxacillin ^d^	Penicillin ^d^	SCC*mec* Type	*psm-mec* ^e^	PSM-mec ^f^	D-Toxin ^g^
11-025	Surgical	This study	SAMN38979304	84	I	4/14/08	+ ^h^	>2	>8	NA45	- ^i^	-	-
11-033	Pyoderma	This study	SAMN38979305	84	I	5/19/08	+	>2	>8	NA45	-	-	+
18-007	Surgical	This study	SAMN38979306	84	I	12/15/09	+	>2	>8	NA45	-	-	+
29-086	Pyoderma	This study	SAMN38979307	84	I	11/24/10	+	>2	>8	NA45	-	-	+
31-086	Urine	This study	SAMN38979308	84	I	3/22/11	+	0.5	8	NA45	-	-	+
11-041	Pyoderma	This study	SAMN38979297	64	II	6/9/08	+	>2	>8	NA45*^int^*	-	-	+
29-036	Pyoderma	This study	SAMN38979298	64	II	10/29/10	+	>2	>8	NA45*^int^*	-	-	+
30-027	Healthy	This study	SAMN38979299	64	II	12/16/10	+	2	>8	NA45*^int^*	-	-	+
36-067	Surgical	This study	SAMN38979300	64	II	9/9/11	+	2	>8	NA45	-	-	+
42-072	Urine	This study	SAMN38979301	64	II	1/3/13	+	2	>8	NA45*^int^*	-	-	+
11-092	Surgical	[24]	283734.1320	71	III	10/28/08	+	>2	>8	III_(KM1381)_	+	-	-
16-041	Urine	[24]	283734.1335	71	III	10/14/09	+	>2	>8	III_(KM1381)_	+	+	+
16-047	Surgical	[24]	283734.1336	71	III	10/16/09	+	>2	>8	III_(KM1381)_	+	+	+
27-080	Surgical	[24]	283734.1356	71	III	8/19/10	+	>2	>8	III_(KM1381)_	+	-	-
28-009	Surgical	This study	SAMN38979303	71	III	9/1/10	+	>2	>8	III_(KM1381)_	+	+	+
38-020	Surgical	[24]	283734.1438	71	III	11/21/11	+	>2	>8	III_(KM1381)_	+	+	+
39-094	Pyoderma	[24]	283734.1458	71	III	3/16/12	+	>2	>8	III_(KM1381)_	+	+	+
10-098	Pyoderma	[24]	283734.1315	68	IV	2/26/08	+	0.5	>8	V_T_	-	-	+
11-012	Pyoderma	[24]	283734.1316	68	IV	3/17/08	+	>2	>8	V_T_	-	-	+
13-061	Surgical	[24]	283734.1330	68	IV	12/30/08	+	0.5	>8	V_T_	-	-	+
16-021	Urine	[24]	283734.1334	68	IV	9/21/09	+	2	>8	V_T_	-	-	+
19-007	Surgical	This study	SAMN38979302	68	IV	2/2/10	+	0.5	>8	V_T_	-	-	+
22-078	Urine	[24]	283734.1344	68	IV	3/2/10	+	1	>8	V_T_	-	-	+
27-010	Surgical	[24]	283734.1350	68	IV	7/14/10	+	1	>8	V_T_	-	-	+
27-023	Urine	[24]	283734.1352	68	IV	7/20/10	+	0.5	>8	V_T_	-	-	+
32-003	Surgical	[24]	283734.1385	68	IV	3/29/11	+	>2	>8	V_T_	-	-	+
33-021	Surgical	[24]	283734.1393	68	IV	4/28/11	+	>2	>8	V_T_	-	-	+
34-031	Urine	[24]	283734.1401	68	IV	6/16/11	+	2	>8	V_T_	-	-	+
39-002	Urine	[24]	283734.1452	68	IV	1/6/12	+	0.5	>8	V_T_	-	-	+
37-032	Pyoderma	[24]	283734.1401	850	II	10/6/11	-	≤0.25	0.5	NA	-	-	+^f^
32-012	Healthy	[24]	283734.1452	871	II	4/1/11	-	≤0.25	0.5	NA	-	-	+
ED99	Pyoderma	[24]	CP002478.1	25	III	N/A	-	≤0.25	1	NA	-	-	+

^a^ MLST (multilocus sequence types) as determined previously [24]. ^b^ Agr group as determined previously [24]. ^c^ Presence of *mecA* was previously determined by WGS [24] and confirmed by real-time PCR (this study). ^d^ Determined by MIC (µg/mL). ^e^
*psm-mec* gene sequence detected from WGS data. ^f^ Detected in stationary-phase (16 h) cultures through RP-HPLC/ESI-MS. ^g^ This strain produces a δ-toxin variant as described in Maali et al. [30]. ^h^ Present. ^i^ Absent/below detection limit.

## Data Availability

All data are contained within the article.

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
