# Peer review of "Methicillin Resistance Elements in the Canine Pathogen Staphylococcus pseudintermedius and Their Association with the Peptide Toxin PSM-mec"

_antibiotics, 2024, doi:10.3390/antibiotics13020130_

Round 1

Reviewer 1 Report

Comments and Suggestions for Authors

This study uses WGS based analysis to characterize the SCCmec elements of MRSP isolates. This manuscript presents some useful findings which add to the growing understanding of this important canine pathogen. Overall, the experiments have been well conducted and the analysis well performed however, there is discordance between the superscript’s symbols in Table I and the symbols in the footnotes. Additionally, I would recommend that Table I and the Supplementary Table S1 should be combined into a single Table.

Author Response

… however, there is discordance between the superscript’s symbols in Table I and the symbols in the footnotes. Additionally, I would recommend that Table I and the Supplementary Table S1 should be combined into a single Table. 

Reply: ok, corrected and changed as indicated.

Reviewer 2 Report

Comments and Suggestions for Authors

The manuscript by Cheung et al is a meticulously written scientific article. The topic is of wide importance and interest to the researchers working in the field of AMR. All the methodologies employed are state-of-the-art and technically sound. Some minor comments are:

1. the authors have used the abbreviation Agr for antibiotic resistance genes. this should be replaced by ARGs throughout the manuscript.

2. The methodology of ONT sequencing can be elaborated example the kit used.

3. Authors are advised to provide the version of Flye, Medaka Plasmidsaurus Bakta software used for analysis.

4. What were the positive and negative controls used in real-time PCR

Comments on the Quality of English Language

English is fine.

Author Response

  1. the authors have used the abbreviation Agr for antibiotic resistance genes. this should be replaced by ARGs throughout the manuscript.

Reply: Agr does not stand for antibiotic resistance genes, it stands for accessory gene regulator, as explained at first use of the term on line 81.

  1. The methodology of ONT sequencing can be elaborated example the kit used.

Reply: This methodology was now described in detail.

  1. Authors are advised to provide the version of Flye, Medaka Plasmidsaurus Bakta software used for analysis.

Reply: Versions are now included.

  1. What were the positive and negative controls used in real-time PCR

Reply: Positive and negative controls are now described.

Reviewer 3 Report

Comments and Suggestions for Authors

In this paper, the authors characterized Staphylococcal Chromosome Cassette mec (SCCmec) from different MRSP isolates and measured the production of PSM-mec in some specific isolates through bioinformatic and biochemical analyses. This is a well-organized paper. I have some comments and questions for the paper.

Line 23 and 31, it’s kind of difficult to get that SCCmec elements may affect DNA uptake in the manuscript. The authors may consider removing “DNA uptake” or providing more evidence.

In Table 1, it’s better to give the definition of “+” and “-” in the table legends.

In Table 1, for isolates 11-092 and 27-080, it looks like that the results of psm-mec gene sequence detected from WGS data and PSM-mec by RP-HPLC/ESI-MS are different, could authors give some explanations?

In Figure 3, for the production of PSM-mec, there are no error bars in the bar graphs. Is that only one-time experiment results? 

Author Response

Line 23 and 31, it’s kind of difficult to get that SCCmec elements may affect DNA uptake in the manuscript. The authors may consider removing “DNA uptake” or providing more evidence.

Reply: What is meant in the abstract is explained in the sentences following this statement (R-M systems and CRISPR may impact DNA uptake and psm-mec virulence.) This is further detailed and explained in the manuscript (results and discussion).

In Table 1, it’s better to give the definition of “+” and “-” in the table legends.

Reply: “+” and “-“ are now explained in the footnote.

In Table 1, for isolates 11-092 and 27-080, it looks like that the results of psm-mec gene sequence detected from WGS data and PSM-mec by RP-HPLC/ESI-MS are different, could authors give some explanations?

Reply: See lines 254-258 where this has been addressed and explained. Detection of delta-toxin was performed specifically to answer this question.

In Figure 3, for the production of PSM-mec, there are no error bars in the bar graphs. Is that only one-time experiment results? 

Reply: Yes, these are single measurements. Please note that for every isolate, an RP-HPLC run was necessary. To do this in triplicate would require a very high number of runs that we did not deem necessary as in our experience the variation is very low for this sort of analysis that we performed many times over many years in our lab.